# Cost-utility analysis of Cryoballoon ablation versus Radiofrequency ablation in the treatment of paroxysmal atrial fibrillation in Iran

**Ali Darvishi[1,2], Parham Sadeghipour[3], Alireza Darrudi[2], Rajabali Daroudi[2]***

**1** Chronic Diseases Research Center, Endocrinology and Metabolism Population Sciences Institute, Tehran University of Medical Sciences, Tehran, Iran, **2** Department of Health Management and Economics, School of Public Health, Tehran University of Medical Sciences (TUMS), Tehran, Iran, **3** Cardiovascular Intervention Research Center, Rajaie Cardiovascular Medical and Research Center, Iran University of Medical Sciences, Tehran, Iran

* rdaroudi@yahoo.com

## Abstract

Atrial fibrillation (AF) is the most prevalent cardiac arrhythmia (Calkins H, et al. 2012). There are various methods to treat AF of which Ablation is one of the most effective. We aimed to assess the cost-utility of Cryoballoon ablation (CBA) compared to Radiofrequency ablation (RFA) to treat patients with paroxysmal AF in Iran. A cost-utility analysis was done using a decision-analytic model based on a lifetime Markov structure which was drawn considering the nature of interventions and the natural progress of the disease. Costs data were extracted from medical records of 47 patients of Shahid Rajaie Cardiovascular Medical Center in Tehran in 2019. Parameters and variables such as transition probabilities, risks related to side effects, mortality rates, and utility values were extracted from the available evidence. Deterministic and probabilistic sensitivity analysis was also done. TreeAge pro-2020 software was used in all stages of analysis. In the base case analysis, the CBA strategy was associated with higher cost and effectiveness than RFA, and the incremental cost-effectiveness ratio was $11,223 per Quality-adjusted life year (QALY), which compared to Iran's GDP per capita as Willingness to pay threshold, CBA was not cost-effective. On the other hand, considering twice the GDP per capita as a threshold, CBA was cost-effective. Probabilistic sensitivity analysis confirmed the findings of base case analysis, showed that RFA was cost-effective and the probability of cost-effectiveness was 59%. One-way sensitivity analysis showed that the results of the study have the highest sensitivity to changes in the RFA cost variable. Results of sensitivity analysis showed that the cost-effectiveness results were not robust and are sensitive to changes in variables changes. Primary results showed that CBA compared to RFA is not cost-effective in the treatment of AF considering one GDP per capita. But the sensitivity analysis results showed considerable sensitivity to changes of the ablation costs variable.

**Data Availability Statement:** All relevant data are within the paper and its Supporting Information files.

**Funding:** The author(s) received no specific funding for this work.

**Competing interests:** The authors have declared that no competing interests exist.

# Introduction

Atrial fibrillation (AF) is the most prevalent cardiac arrhythmia in which electrical stimulation does not follow a definite pathway. It occurs when an electrical wave has no distinct direct in atria and is described as supraventricular tachyarrhythmia which is accompanied by uncoordinated atrial activity and subsequently atrial mechanical failure [1].

Stages of AF are based on the duration of arrhythmia which is classified into three paroxysmal, persistent and permanent stages. At a paroxysmal stage, the arrhythmia period is more than 30 seconds and shorter than one week, and if arrhythmia lasts for more than 7 days and lesser than one year, the disease will develop in a persistent stage, and finally, if arrhythmia lasts for more than one year, the disease entered into permanent stage [2].

AF affected 21 million men and 13 million women based on 2010 data which prevalence rate is much more in developed countries [3, 4]. According to present evidence, about one-third of cardiac arrhythmia hospitalizations are due to AF and its rate has increased up to 66% over the last 20 years. This increase can be due to the process of aging of the population, an increase in the prevalence of cardiac chronic diseases, and an increase in diagnosis cases due to advances in diagnostic technologies [5].

There are various methods to treat AF, and catheter ablation is one of the most important methods. Ablation is a non-surgical method that removes the region which consists of abnormal pathways with specific waves. Nowadays this method is widely used to treat types of atrial tachycardia (rapid pulse rate), such as AF, atrial flutter, and some types of ventricular tachycardia. In this method, an electrophysiologist inserts into heart cavities one or more catheters with electrodes at the end and uses a type of energy to ablate the abnormal texture of the heart, which causes extra electrical messages. The area of heart tissue that is ablated is too small and does not affect the total function of the heart. A small and safe repaired tissue in this area is formed and the normal rhythm of the heart will return [6, 7].

Ablation has various types. One type is point-by-point ablation around vessels using Radiofrequency ablation (RFA). In this method, a wire is entered through the groin, and the focal point of arrhythmia is burnt by entering these waves [7]. Another type is Cryoballoon ablation (CBA). In this method, the physician enters a wire into the heart through the groin and places it at the focal point of arrhythmia. But this balloon is cooled through nitrogen flow, and the focal point of arrhythmia is ablated through freezing. In uncoordinated arrhythmia of AF which has numerous focal points of the arrhythmia, one type of balloon could be used which spontaneously the focal points are frozen by nitrogen flow [7].

According to recent findings, ablation technologies are the most effective therapeutic methods to improve the status of patients with AF and have the highest effect on preserving cardiac sinus rhythm as well as improving quality of life [8, 9], but besides, also have a different financial burden and risk load rather than other therapeutic methods.

There are some evidence regarding differences between two ablation technologies which some of these studies showed that these two methods have identical clinical efficacy and safety, have almost low side effects and showed no considerable preference in none of the methods [10, 11]. But economically, studies around the world showed sometimes contradictory results regarding the cost-effectiveness of each method. For example, the results of the study by Ming et al. (2019) in china and Murray et al. study (2019) in Germany showed that CBA is a cost-effective strategy compared to RFA. In contrast, in the study by Sun et al. (2019), the results showed the cost-effectiveness of RFA.

In Iran, because CBA is a relatively new technology in Iran, studies on comparing the costs and effectiveness related to this technology with other existing treatment methods are scarce,

and based on our information, no full economic evaluation study has been conducted to compare these two technologies, and the present study is the first study in Iran in this regard.

To provide appropriate evidence to decide on application and coverage of the most appropriate technology, the current study was designed to assess the cost-utility of CBA compared to RFA to treat patients with paroxysmal AF in Iran.

## Materials and methods

The present study is a full economic evaluation based on a decision-analytic model which compared two strategies of CBA and RFA in Iran. Accordingly, the cost-utility analysis method was used. Various stages of economic evaluation were performed based on reference guideline of the national institute for health and care excellence (NICE) to perform economic evaluation of health technologies [12]. Also, we matched the study reports and findings with the CHEERS checklist [13].

### Modeling

The economic model was designed based on a literature review, the natural history of the disease, the process of performing ablation methods in patients with paroxysmal AF, clinical outcomes, probabilities of occurrence of outcomes, and incidence of expenses.

To design the model, specialized panels were formed with the presence of a team of clinical specialists and economic team. After trial and error of various models, the final model was extracted by consensus and considering the most important clinical and economical outcomes based on natural history. The designed economic evaluation model is demonstrated in Fig 1. The structure is the same for both arm.

The model structure is designed so that each strategy is based on a lifetime Markov structure with a one-year cycle length. In each Markov structure, five health states including AF pre-intervention, normal sinus rhythm (NSR), AF post-intervention, AF post-re-intervention, post-stroke, and death were considered. Individuals in both comparable groups were at the AF before ablation health state in zero cycle. It was assumed that individuals in each health state remained at the end of each cycle or went through other health states or died. Also, patients can place in are-ablation state once, and due to lack of sufficient evidence, third ablations and higher were not considered in the model, and considering AF post-re-intervention health state was for this purpose.

To extract evidence regarding mortality rate and other evidence, the mean age of 50 years (due to the mean age of patients undergoing ablation based on accessible hospital information from Shahid Rajaie Cardiovascular Medical and Research Center in Tehran) for patients at initiation of Markov models were considered.

**Extracting parameters and analyzes.** This study was performed from the perspective of Iran's health system and as mentioned, the time horizon was considered as lifetime, and costs and outcomes were estimated accordingly.

Since the current study was a full economic evaluation, the quality-adjusted life years (QALYs) index was considered as outcome measure and its value at each health state is computed by estimating patients' utility at each state. Also, in this section, the amount of disutility caused by the side effects of ablation in health states was considered.

The Status of cost-effectiveness of each strategy was finally assessed based on cost per unit of QALY. Evidence related to patients' utility at each health state was extracted from international studies. Regarding costs, due to the study perspective, direct medical costs were only considered.

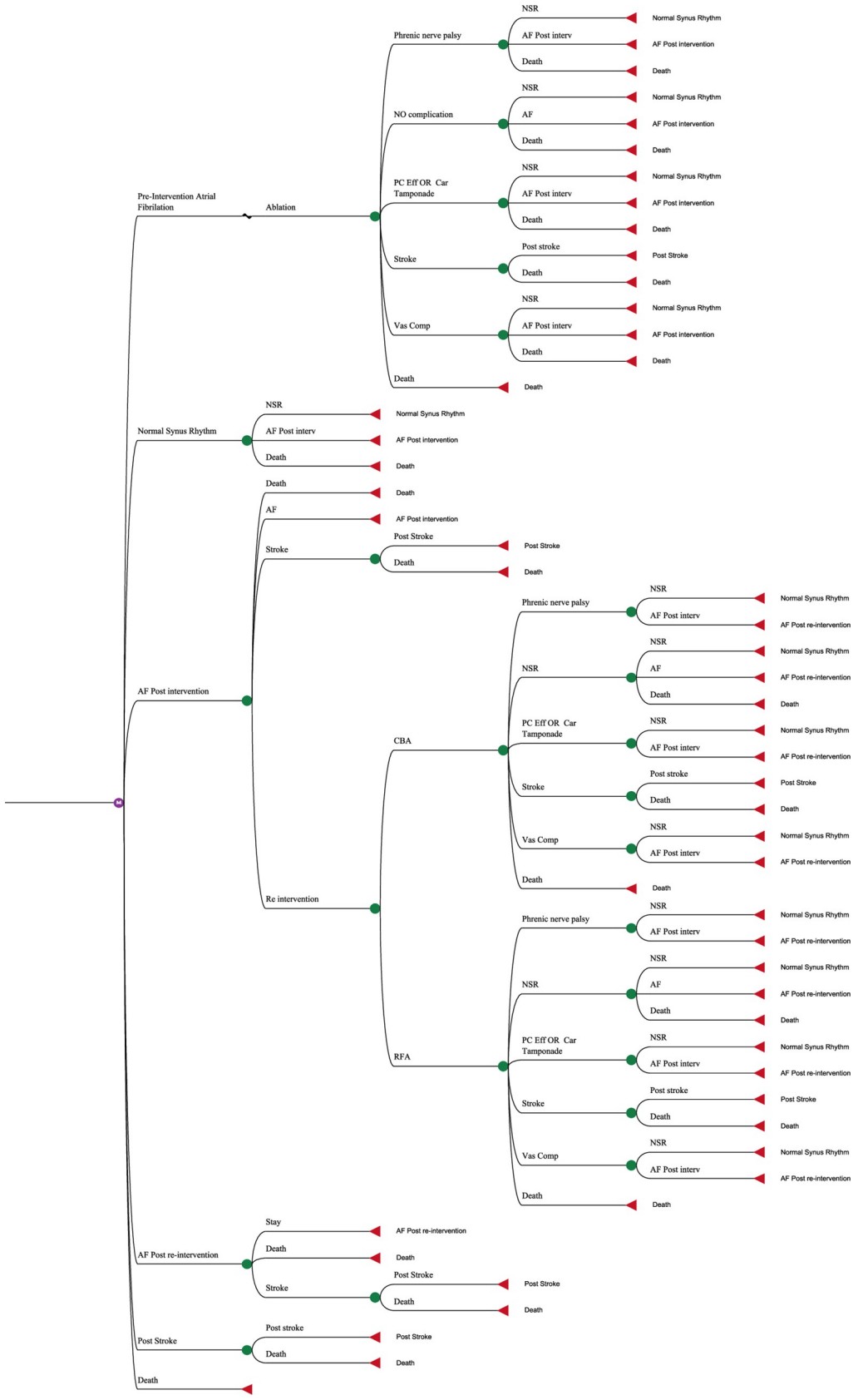

**Fig 1. Markov model for CUA of CBA vs RFA.**

The cost of each strategy was estimated by the cost of various health states based on cost units used in interventions. Cost of performing the procedure of CBA and RFA including the cost of CBA and RFA, costs related to management and supervision of receiving services in hospital, costs of hospitalization, supportive, therapeutic, and pharmaceutical cares, and costs related to side effects were considered. Data collection for costing was from medical records of 47 patients of Shahid Rajaie Cardiovascular Medical and Research Center in Tehran in 2019–20. Since all the medical records related to 2019–20 were accessible, sampling was not done for this purpose, and hospital bills of all patients undergoing both ablation methods were assessed. Accordingly, 27 and 20 medical records related to CBA and RFA were assessed, respectively. The cost of other medical attempts required at health states independent from the cost of the procedure was determined based on therapeutic protocols. Accordingly, the type of anti-coagulant and anti-arrhythmic medications, and their doses until a definite time, cost of hiring a Holter monitor device, and cost of electrical cardioversion were determined. All the mentioned cases were different in various patients and therefore, by consultation with clinical specialists, a moderate amount of costs was considered. The cost of side effects was estimated based on evidence of previous studies and re-costing based on tariffs and domestic currency (Table 1).

All the stages of costing were calculated by holding a specialized panel with a clinical team and based on governmental tariffs of Iran's Ministry of Health. All costs were calculated based on 2019–20 prices.

Other parameters and variables related to transition probabilities among health states of two strategies, the risk of mortality at each health state, mortality risk caused by ablation methods, the efficacy of interventions, and other parameters including the risk of side effects of each ablation method were extracted from international evidence. In this regard, based on each parameter, a distinct literature review was performed in scientific databases, and studies with appropriate evidence were classified and finally, the best evidence was extracted. In terms of side effects, we included "pericardial effusion", "cardiac tamponade", "permanent phrenic nerve palsy", "vascular complications" and "Stroke" as side effects of the technologies in the model, which we considered both their related cost and the disutility values. The choice of side effects in the present study was based on the evidence from previous studies and also in consultation with the clinical specialists. Values of parameters and variables of the model and their references are given in Table 1.

We used a 5% the discount rate for both costs and QALYs in the model based on the recommendation of the Health Technology Assessment Office of Iran's Ministry of Health for this study.

## Cost-effectiveness analysis

To perform analysis and determine the most cost-effective strategy, due to the costs and effectiveness of each strategy, the incremental cost-effectiveness ratio (ICER) was calculated.

The equation for this index was as below:

$$ICER = C_1 - C_2 / E_1 - E_2$$

In, $C_{1, 2}$ represents the cost of CBA and RFA, and $E_{1, 2}$ represents their effectiveness.

In this study, the cost-effectiveness threshold was considered to be one time of GDP per capita equal to $7142. This choice was made on the recommendation of the World Health Organization to select 1 to 3 times the per capita GDP in developing countries [25].

To perform all stages of modeling and analysis of results, TreeAge 2020 software was used.

**Table 1. Model inputs and parameters.**

| Statistic variable | Base case | SD/(CI) | Distribution | Source |
|---|---|---|---|---|
| Annual discount rate (Costs and QALYs) | 0.05 | (0–0.1) | Beta | |
| Time Horizon(year) | Life Time | | | |
| *Transition Probabilities (CBA group)* | | | | |
| AF recurrence after ablation, first year | 0.269 | ±0.0538 | Beta | [14] |
| AF recurrence after ablation, >first year | 0.0938 | ±0.0235 | Beta | [15] |
| Re-intervention with RFA | 0.5516 | ±0.11032 | Beta | [15] |
| Re-intervention with CBA | 0..0951 | ±0.01902 | Beta | [15] |
| *Probability of Complications (CBA group)* | | | | |
| pericardial effusion or cardiac tamponade | 0.0084 | | | [14] |
| permanent phrenic nerve palsy | 0.032 | | | [14] |
| vascular complications | 0.0156 | | | [14] |
| Stroke rate per year | 0.05 | | | [16] |
| *Transition Probabilities (RFA group)* | | | | |
| AF recurrence after ablation, first year | 0.3326 | ±0.0665 | | [14] |
| AF recurrence after ablation, >first year | 0.1055 | ±0.0264 | | [15] |
| Re-intervention with RFA | 0.5685 | ±0.1137 | | [15] |
| Re-intervention with CBA | 0.0587 | ±0.01174 | | [15] |
| *Probability of Complications (RFA group)* | | | | |
| pericardial effusion or cardiac tamponade | 0.0231 | | | [14] |
| permanent phrenic nerve palsy | 0.0005 | | | [14] |
| vascular complications | 0.023 | | | [14] |
| Stroke per year | 0.05 | | | [16, 17] |
| *Probability of death* | | | | |
| Annually probability of death of 50 yrs Normal Population (First year)* | 0.0037 | | | Iran Life Table [18] |
| Probability of operative death | 0.000487 | | | [19] |
| Stroke-specific mortality | 0.3536 | | | [20] |
| *Costs($)* | | | | |
| AF average annual costs | 372.81 | ±55.92 | Gamma | Our Study |
| NSR average annual costs | 273.32 | ±40.99 | Gamma | Our Study |
| CBA | 7751.88 | ±516.72 | Gamma | Our Study |
| RFA | 5027.10 | ±1530.66 | Gamma | Our Study |
| Stroke costs, first year | 1804.49 | ±180.44 | Gamma | Our Study |
| Post-stroke costs, >first year | 541.34 | ±54.13 | Gamma | Our Study |
| pericardial effusion or cardiac tamponade | 1060.19 | ±106.01 | Gamma | [9], adjustment |
| permanent phrenic nerve palsy | 11.09 | ±106.01 | | [9], adjustment |
| vascular complications | 60.23 | | | [9], adjustment |
| *Utilities* | | | | |
| NSR | 0.8 | ±0.00577 | Beta | [21] |
| AF | 0.6 | ±0.0721 | Beta | [21] |
| Post-stroke | 0.46 | ±0.0577 | Beta | [22] |
| Disutility due to complications | -0.0314 | | Beta | [23] |
| Disutility due to stroke, first year | -0.296 | | Beta | [24] |

* A complete table of Probability of death at different ages is given as S1 Table.

## Sensitivity analysis

Given the uncertainty regarding some parameters used in the model, Deterministic and Probabilistic Sensitivity Analysis of the results of the model was performed.

To perform Deterministic Sensitivity Analysis (DSA), one-way sensitivity analysis, and Tornado diagram, and two-way sensitivity analysis were used.

PSA was performed considering the probability distribution of uncertain variables using Monte Carlo simulation. The range used for uncertainty in point estimation of each variable and statistical distributions used in PSA is presented in Table 1. For example, for the cost variables gamma distribution and other variables such as the transition probabilities, the beta distribution was considered. In cases in which no evidence regarding the variance of the variable was found, 10% or 20% of the mean values of variable were considered as standard deviation, and appropriate distribution was selected due to the type of variable.

**Ethics approval and consent to participate.**   The Present study did not make use of human or animal subjects and/or tissue.

This research was approved by the ethical committee of Tehran University of Medical Sciences (TUMS) by the code of IR.TUMS.VCR.REC.1399.453.

# Results

## Base case analysis

Table 2 represents the results of the cost-utility analysis. Accordingly, the results showed that average lifetime costs per patient associated with CBA and RFA were $14,198 and $12,005, respectively. Average QALYs per patient were estimated at 8.469 and 8.273 for both strategies, respectively. Accordingly, ICER was estimated at $11,223.85 per QALY unit, which compare to Iran's WTP threshold represents a lack of cost-effectiveness of CBA technology compare to RFA.

Fig 2 also showed the cost-effectiveness plane of analysis. As observed, the CBA strategy is associated with higher costs and higher effectiveness than the RFA, and this amount of QALYs obtained in exchange for the increased cost due to CBA is not cost-effective based on the considered threshold.

## Sensitivity analysis

**Deterministic Sensitivity Analysis (DSA).**   One-way DSA of all uncertain variables is presented in Fig 3 using the Tornado diagram. Variables include the cost of CBA, cost of RFA, probability of recurrence after CBA, and RFA and probability of re-ablation using CBA in the RFA group were considered for DSA. As observed in Fig 3, changes in values of RFA cost had the highest effect, and the probability of recurrence after CBA had the lowest effect on the results of the study. Besides, based on the diagram, all uncertain variables except for recurrence after RFA consist threshold and in distinct values change results of final analysis according to

**Table 2. Results of base case cost-effectiveness analysis.**

| Strategy | RFA | CBA |
|---|---|---|
| Cost($) | 12,005.20 | 14,198.36 |
| QALYs | 8.273 | 8.469 |
| Incremental Cost($) | - | 2,193.16 |
| Incremental QALYs | - | 0.195 |
| ICER($/QALY) | - | 11,223.85 |

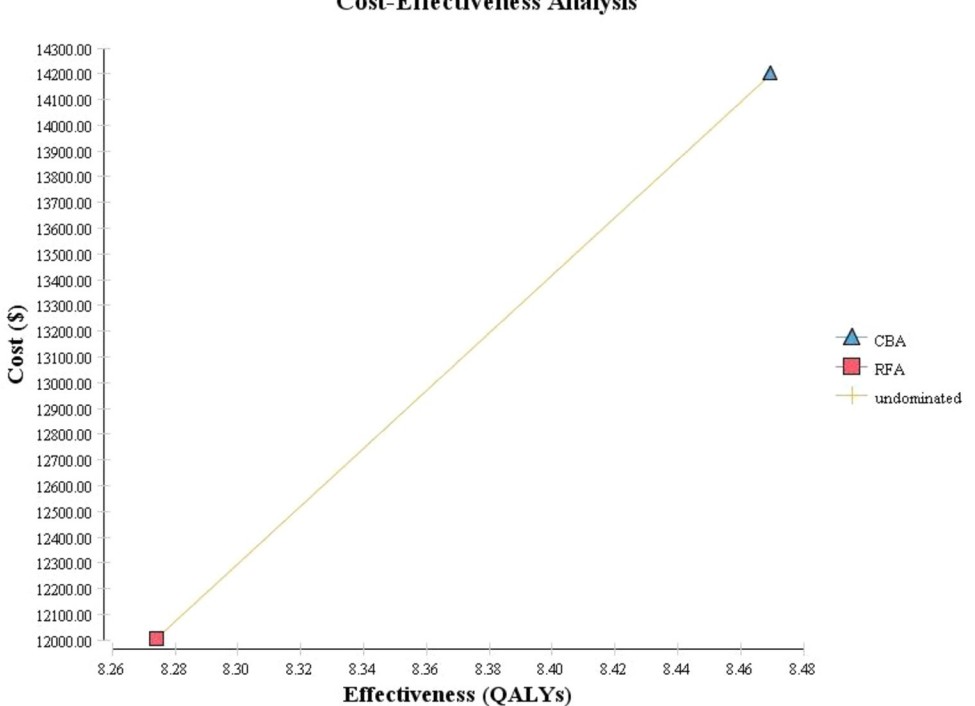

**Fig 2. Cost-effectiveness analysis of CBA vs RFA.**

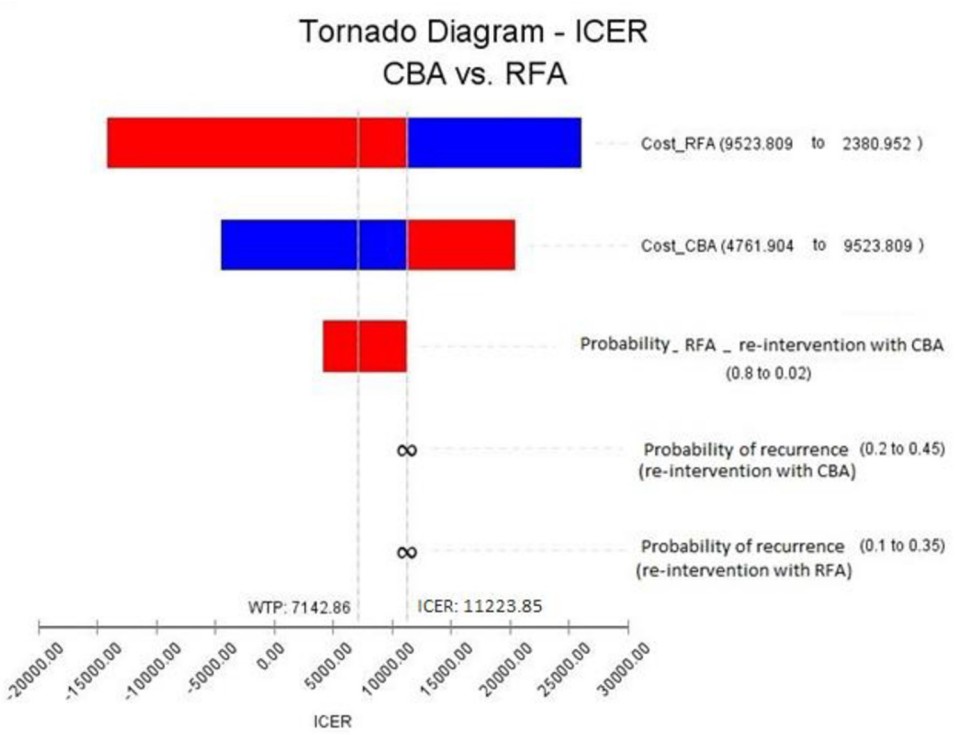

**Fig 3. One way sensitivity analysis using Tornado diagram.**

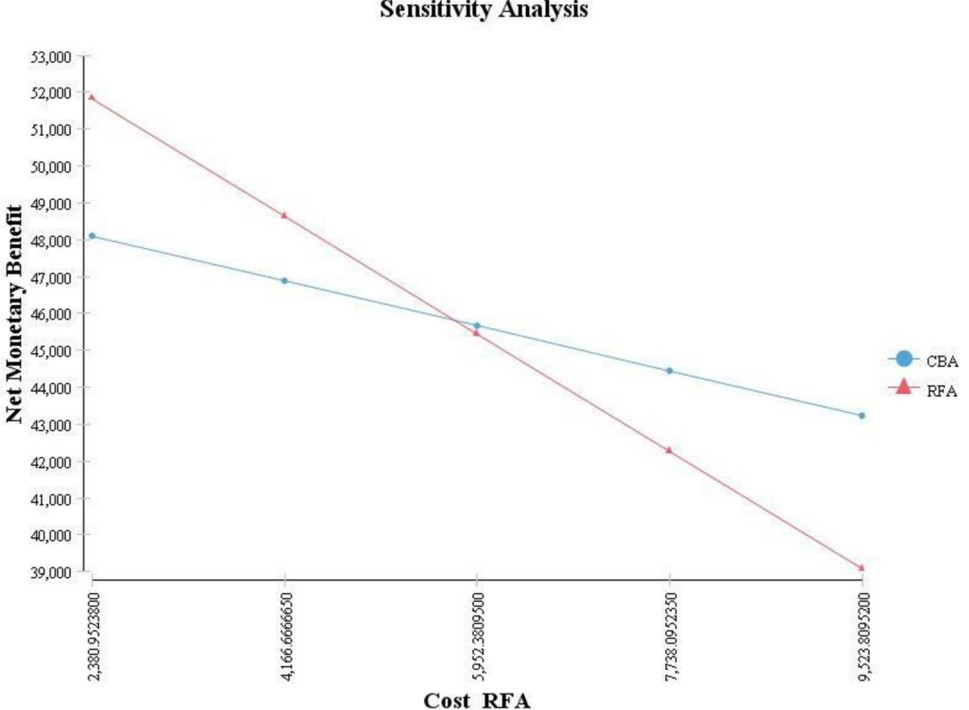

**Fig 4. One way sensitivity analysis (RFA cost).**

the considered confidence interval. It should be noted that to assess the uncertainty of parameters more precisely, the confidence interval of variables was considered widely.

Fig 4 shows the results of one-way sensitivity analysis of RFA cost. As observed, this variable consists of the threshold at the value of $5738 (about $714 higher than the base case), and at this value, the results of the cost-utility analysis were changed and showed approximately high sensitivity to changes in this variable. To assess more precisely, two-way sensitivity analysis was done based on the changes in the cost of CBA and RFA variables. As observed in Fig 5, the results of the cost-utility analysis show sensitivity to changes in these two variables.

In general, the results of DSA showed that the results of analysis have considerable sensitivity to changes in uncertain variables.

**Probabilistic Sensitivity Analysis (PSA).** By considering the function of the probability distribution of uncertain variables, PSA was done using Monte Carlo simulation by considering a number of the 10,000 times of repeating simulation and sampling.

Fig 6 shows that Incremental Cost-Effectiveness scatter plot of CBA versus RFA in 10000 times of repeating sampling and simulation. In addition, Table 3 presented a report on the probability of placement of the CBA strategy at each cost-effectiveness plot region compared to the RFA strategy. As it is observed, the probability of placement of CBA at regions of I, III, and IV and below the WTP threshold (cost-effectiveness regions) was 41%, and the probability of cost-effectiveness of RFA was 59% which these results, confirmed base case analysis results.

## Discussion

This study aimed to analyze the comparative cost-utility of two technologies of CBA and RFA in the treatment of patients with paroxysmal AF in Iran.

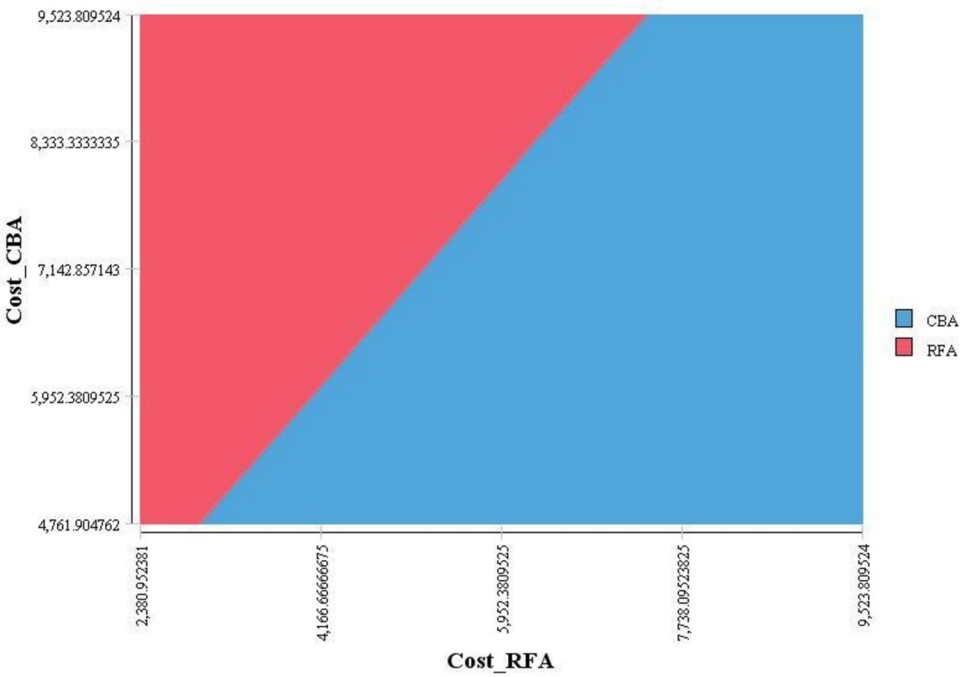

**Fig 5. Two way sensitivity analysis (RFA cost and CBA cost).**

**Fig 6. Incremental cost-effectiveness scatter plot for PSA.**

**Table 3. Report of the cost-effectiveness probability in each cost-effectiveness plane quadrants.**

| Component | Quadrant | Incre Eff(IE)* | Incre Cost(IC)** | Incre CE*** | Frequency | Proportion (%) |
|---|---|---|---|---|---|---|
| C1 | IV | IE>0 | IC<0 | Superior | 1,388 | 0.1388 |
| C2 | I | IE>0 | IC>0 | ICER<3.0E8 | 2,772 | 0.2772 |
| C3 | III | IE<0 | IC<0 | ICER>3.0E8 | 35 | 0.0035 |
| C4 | I | IE>0 | IC>0 | ICER>3.0E8 | 3,103 | 0.3103 |
| C5 | III | IE<0 | IC<0 | ICER<3.0E8 | 41 | 0.0041 |
| C6 | II | IE<0 | IC>0 | Inferior | 2,661 | 0.2661 |
| Indiff | origin | IE = 0 | IC = 0 | 0/0 | 0 | 0 |

*Incremental Effectiveness

**Incremental Cost

***Incremental Cost Effectiveness

Evidence review of these two novel technologies showed that in most countries they have better safety and efficacy rather than anti-arrhythmia medications in returning NSR [26–28]. Regarding the clinical efficacy of two technologies in the treatment of AF patients, most review studies showed that there is no significant difference between the two methods as well as their side effects, and in this regard, none of them is superior [7, 10, 29, 30]. Of course, some other studies show a little superiority of CBA in this regard [14]. Therefore, it seems that economic assessment of these two technologies through applying them in the health care system is of most importance. Results of the present study showed that in a base case analysis of CBA and RFA comparison, CBA at WTP threshold of $7,142 was not cost-effective, but was cost-effective at a WTP of $14,285 (twice the GDP per capita).

Previous studies comparing these two technologies in various countries also show controversial results. For example, studies by Murray et al. (2018) in Germany and Sun et al. (2019) in china confirmed our findings [9, 31], in contrast, a study by Ming et al. (2019) in China showed that CBA compared to RFA is the dominant strategy in the treatment of patients with AF due to lower cost and QALY values [15].

PSA of results also confirmed findings of base case analysis, and accordingly, results showed that by repeating sampling and simulation based on statistical distributions of uncertain variables, RFA strategy by the probability of about 60% would be cost-effective. DSA showed that results of analysis have a higher sensitivity to changes of some variables, and in some values, variables have a threshold and could change total results. Results of the study had the highest sensitivity to changes in RFA costs. Accordingly, as observed, if the cost of the RFA is $5,738 instead of $5,127, the results will change and CBA will be cost-effective.

By more assessment on variables of RFA and CBA costs and considerable difference in cost of these two technologies, it was clarified that a part of this cost difference might be due to re-use of some procedure requirements and not applying some sidelong tools in the procedure of RFA which decreases costs of this intervention. This issue could increase the risk of side effects and decrease the efficacy of treatment considering comments of specialists, but since no appropriate evidence was found for this issue, it was not considered in the economic evaluation model. In other words, decreasing in costs was considered in the model due to re-use of procedure requirements, but negative outcomes related to that were not considered due to lack of appropriate evidence, which this issue increased the chance of RFA for cost-effectiveness. Accordingly, it seems that the results of the study must be interpreted and used cautiously.

The present study is the first economic evaluation for the comparison of ablation technologies in Iran. As mentioned, just hospital data of a specialized hospital was used, and due to lack

of access to hospital data of other centers, a comparison of costs at various centers was not achieved. Besides, since clinical studies regarding the efficacy of technologies and associated side effects in Iran were not found, the best present evidence of international studies was just used. Although in many cases there is no significant difference between different contexts regarding clinical evidence, this is one of the possible limitations of this study that can be effective in the final results. However, to eliminate this possible limitation as much as possible, in the present study we performed a sensitivity analysis of a wide range for uncertain parameters. Another limitation of the study was regarding re-ablations. So that, appropriate evidence-based on substitution of each technology in case of failure of primary ablation was not found in Iran, and thereby, international evidence was also used in this regard.

## Conclusions

Findings obtained from our study showed that based on Iran's Health system perspective, CBA technology compared to RFA is not a cost-effective strategy to treat patients with paroxysmal AF at a threshold of one time of Iran's GDP per capita. This is while, considering twice the GDP per capita and higher as the threshold, CBA was cost-effective. On the other hand, the results of sensitivity analysis showed that results of the evaluation model have considerable sensitivity to changes in uncertain variables such as ablation costs. In general, it is not possible to conclude with certainty about the cost-effectiveness of CBA against RFA in Iran.

## Supporting information

**S1 Table. Probability of death at different ages (Iran Life Table).**
(DOCX)

## Author Contributions

**Conceptualization:** Parham Sadeghipour, Rajabali Daroudi.

**Data curation:** Ali Darvishi, Alireza Darrudi.

**Formal analysis:** Ali Darvishi, Rajabali Daroudi.

**Investigation:** Ali Darvishi, Alireza Darrudi.

**Methodology:** Ali Darvishi, Rajabali Daroudi.

**Project administration:** Rajabali Daroudi.

**Software:** Ali Darvishi.

**Writing – original draft:** Ali Darvishi, Rajabali Daroudi.

**Writing – review & editing:** Ali Darvishi, Parham Sadeghipour, Alireza Darrudi, Rajabali Daroudi.

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
