## [Decision Letter · Decision Letter 0]

26 Jan 2022

PONE-D-21-27154

Cost-Utility Analysis of Cryoballoon Ablation versus Radiofrequency Ablation in the Treatment of Paroxysmal Atrial Fibrillation; Case Study: Iran

PLOS ONE

Dear Dr. Daroudi,

Thank you for submitting your manuscript to PLOS ONE. After careful consideration, we feel that it has merit but does not fully meet PLOS ONE’s publication criteria as it currently stands. Therefore, we invite you to submit a revised version of the manuscript that addresses the points raised during the review process.

Please ensure the following points are addressed: 

<ul><li> 

Use CHEERS checklist for reporting</li><li> 

Introduction needs to provide more details on what is already known on the economic aspects rather than clinical ones</li><li> 

Model and parameters require more details and justification. Address specific points provided by reviewers </li><li> 

Parameters need more justification and details. Check and justify choice of transition probabilities, distributions, discount rate, threshold</li><li> 

Sensitivity analysis to be repeated using 10000 iterations</li><li> 

Tables and figures to be adjusted</li><li> 

Paper to be proof edited </li></ul>

We look forward to receiving your revised manuscript.

Kind regards,

Elena Pizzo, PhD

Academic Editor

PLOS ONE

https://journals.plos.org/plosone/s/file?id=ba62/PLOSOne_formatting_sample_title_authors_affiliations.pdf”.

a) Did participants provide their written or verbal informed consent to participate in this study?

b) If consent was verbal, please explain i) why written consent was not obtained, ii) how you documented participant consent, and iii) whether the ethics committees/IRB approved this consent procedure."

Reviewers' comments:

Reviewer's Responses to Questions

**Comments to the Author**

1. Is the manuscript technically sound, and do the data support the conclusions?

Reviewer #1: Yes

Reviewer #2: Yes

2. Has the statistical analysis been performed appropriately and rigorously? 

Reviewer #1: N/A

Reviewer #2: Yes

3. Have the authors made all data underlying the findings in their manuscript fully available?

Reviewer #1: Yes

Reviewer #2: Yes

4. Is the manuscript presented in an intelligible fashion and written in standard English?

Reviewer #1: Yes

Reviewer #2: Yes

5. Review Comments to the Author

Reviewer #1: General comments:

This paper focusses on the cost-effectiveness of two alternative treatment methods (Cryoballoon Ablation versus Radiofrequency Ablation) for paroxysmal Atrial Fibrillation. I would like to thank the authors for their work. I think the study is interesting and it can be brought to a publishable level after some revisions.

I think the authors need to use a checklist to ensure a high-quality standard of reporting, such as CHEERS. More details and justification needed when explaining the model and the parameters. I think the probabilistic sensitivity analyses need to be repeated as I have some concerns over the choice of comparator. There are also some inconsistencies between the Figures and the text. I think some the tables need to be re-organised as well. I explain these below.

The text needs to be updated to improve the flow and it should be grammatically checked. The authors may find using online applications (such as Grammarly) helpful. There is repetition at some places, so please make sure there is not two different sentences with the same meaning. Please also add page numbers to the manuscript.

Specific comments:

Page 9 – Abstract:

Line 51. Please provide a reference here.

Line 52. "should be of which"

Line 53. Please add paroxysmal before AF to be clear. I also suggest removing "analysis" and inserting "the" before cost utility.

Line 60. Please remove the word “rather” as it is not used to make comparisons.

Line 61. I think "one time" unnecessary here since when it is just the GDP we understand that it is one time of the GDP.

Line 62. Please remove “but” to avoid redundancy

Lines 63 and 65. It sounds as if PSA and Monte Carlo simulation are two different analyses while in fact, they refer to the same analysis. I suggest re-writing this sentence. Also, instead of saying "by probability of 60%" it might be clearer to say, "the probability of cost-effectiveness was 60%."

Page 10 – Introduction:

I think in the introduction, there is too much focus on the clinical aspects and too little on the cost-effectiveness. The authors need to explain what we already know about the cost-effectiveness of CBA and RFA then tell us about the limitations of existing economic models and why a new model was needed.

Line 73. It is hard to follow this sentence so, I suggest dividing into two or three sentences.

Page 11

Line 115. Please, here and everywhere in the text, add "paroxysmal" to be clear

Page 12

Line 121. This guideline is not for health projects but for health technologies.

Line 124. I think the authors refer to the design of the economic model - I'd update this sentence as “The economic model was designed based on a literature review ...”

Line 130. The word “observed” is not meaningful here it should be replaced with a word like “demonstrated”.

Line 132. For consistency, please consider updating the first state's name to "before intervention" or the word "intervention" in the other states to "ablation".

Lines 136 and 138. Please re-phrase this sentence to be clear and I'd suggest starting it with "It was assumed that..."

Page 13

Figure 1. I think the model demonstration in Figure 1 doesn't match the description in the text.

I think the authors should draw another tree. For example, one of the health states is Atrial Fibrillation I think this should be pre-intervention. Similarly, it is confusing to see CBA just after Atrial Fibrillation but not RFA.

In the methods section, the authors need to explain and justify why specific complications were chosen and not the others.

Additionally, the price year needs to be stated in the methods section.

Line 139. Evidence is a singular word: we cannot write evidence. Please correct this everywhere

Line 140. This needs to be explained more which hospital? The hospital that you took the cost data. You may also cite the existing studies here. Additionally, please rephrase this sentence.

Lines 152 and 153. I don't understand this sentence: could you please rephrase?

Line 163. This needs to be supported with published evidence.

Page 14

Lines 165 and 166. A list of costs which shows the figures taken from the literature and the figures estimated by the authors should be presented as an Appendix.

Line 168. Is it publicly available? If so, please provide a reference or provide a summary table as an Appendix.

Table 1. The authors state following the NICE recommendations; however, they used a discount rate of 0.05. This should be explained and justified.

Table 1 includes mortality during first year, but it is not clear what probability was used after the first year. The authors need to explain this in the text.

Table 1 reports the probabilities for complications under the title of “transition probabilities” and this is really confusing. I think Table 1 needs to be re-organised such that the transition probabilities and morbidity risks are clear.

The SD or CI should be provided for all the probabilities

What is a life table? Please cite a reference or provide this table as an Appendix.

In Table 1 some parameters were identified through “survey and calibration” this needs to be explained in the text. What do you mean when you say survey and calibration? We need to see the original figures/values and those used in the model after adjustments.

Page 16

Line 189. Citation needed here. I think WHO recommends using three times of GDP as the threshold. Please explain why you have chosen to use the GDP. Please see this paper on the thresholds recommended by WHO https://www.who.int/bulletin/volumes/93/2/14-138206.pdf

Line 200. This needs to be more specific. 10%, 20% or a range of values between 10% and 20%?

Line 201. Please provide a reference here to show how you have chosen the appropriate distributions.

Page 17

Line 204. Please use commas for costs to make sure these are clear. For example, is it $14,198.3?

Table 2. Please re-organise this table and report the incremental figures in a separate raw. Also, please change the column name from Eff** to QALYs to be more specific.

Page 18

Figure 3. This figure is confusing because the impact on the net benefit was demonstrated which was not mentioned anywhere in the text. Please repeat this analysis to show the impact of changing inputs on ICER calculations. Additionally, we cannot tell which lower and upper values were used. So, please provide these on the graph. Please re-label the parameters and try not to use abbreviations. Please

also explain what EV represents. Again, use commas for costs.

Figures 4 and 5. Please repeat these analyses using ICERs not net benefits.

Line 242. Why 1000 times? why not 5000 or 10000? This needs to be justified.

Line 243. This sentence raises concern. It was said that the cost-effectiveness of CBA would be estimated compared to RFA. However, the PSA was conducted to estimate the cost-effectiveness of CBA and RFA compared to something else which was not explained to the reader. Is the probability of CBA being cost-effective compared to RFA 59% as stated in the abstract or is the probability of CBA being cost-effective compared to something else is 59%? The authors need to clarify this and repeat the analysis to compare CBA to RFA.

Page 19

Figure 7. This figure is in line with the text, so maybe use this on and remove Figure 6 and related text. Here, please change effectiveness to QALYs.

Table 3. I don't think this makes sense. We don't need this table, just remove it please.

Page 20

Lines 262 and 265. This part should be moved to introduction.

Line 269. Please don't repeat the results here - it would be enough just to say that it was not cost-effective at a WTP of $7,142 but cost-effective at a WTP of $14,285.

Line 273. It would be good to spell out why your results are different than theirs.

Line 279. Why would the cost of RFA increase? Please rephrase this as "the cost of RFA is $5,738 not $5,127 ..."

Line 282. I don't understand this phrase (re-use of necessities). Please explain what this means.

Page 21

Line 298. We wouldn't say viewpoint to describe the perspective in economic evaluations.

Reviewer #2: • Please remove "Case Study" from the title. English language of the manuscript needs to be improved. - Proof-editing the manuscript because grammar, punctuation and use of acronyms is rather poor. Please check keywords with Mesh terms. The authors should harmonize the cost-utility/ cost utility in the manuscript. Lines 188-189: please add some references. Line 190: TreeAge 2011 software/ line 59: TreeAge pro 2020 software. Please correct.

6. PLOS authors have the option to publish the peer review history of their article (what does this mean?). If published, this will include your full peer review and any attached files.

Reviewer #1: **Yes: **Tuba Saygin Avsar

Reviewer #2: No

---

## [Author Response · Author response to Decision Letter 0]

11 Mar 2022

Reviewer 1 Comments:

Dear reviewer

Thank you for your invaluable comments and guidance to improve our manuscript. 

We presented the responses one by one on your comments as follows: 

Reviewer #1: General comments:

This paper focusses on the cost-effectiveness of two alternative treatment methods (Cryoballoon Ablation versus Radiofrequency Ablation) for paroxysmal Atrial Fibrillation. I would like to thank the authors for their work. I think the study is interesting and it can be brought to a publishable level after some revisions.

I think the authors need to use a checklist to ensure a high-quality standard of reporting, such as CHEERS. More details and justification needed when explaining the model and the parameters. I think the probabilistic sensitivity analyses need to be repeated as I have some concerns over the choice of comparator. There are also some inconsistencies between the Figures and the text. I think some the tables need to be re-organised as well. I explain these below.

The text needs to be updated to improve the flow and it should be grammatically checked. The authors may find using online applications (such as Grammarly) helpful. There is repetition at some places, so please make sure there is not two different sentences with the same meaning. Please also add page numbers to the manuscript.

Response: Thanks for your invaluable comments. Final manuscript was revised by native English editor. We matched the study reports and findings with the CHEERS checklist. Also, we applied other specific comments as much as possible.

Specific comments:

Page 9 – Abstract:

Line 51. Please provide a reference here.

Response: It was revised based on your comment.

Line 52. "should be of which"

Response: Thanks for your comment. It was revised based on your comment.

Line 53. Please add paroxysmal before AF to be clear. I also suggest removing "analysis" and inserting "the" before cost utility.

Response: Thanks for your comment. It was revised based on your comment.

Line 60. Please remove the word “rather” as it is not used to make comparisons.

Response: It was removed.

Line 61. I think "one time" unnecessary here since when it is just the GDP we understand that it is one time of the GDP.

Response: Thanks for your comment. It was removed based on your comment.

Line 62. Please remove “but” to avoid redundancy

Response: It was revised based on your comment.

Lines 63 and 65. It sounds as if PSA and Monte Carlo simulation are two different analyses while in fact, they refer to the same analysis. I suggest re-writing this sentence. Also, instead of saying "by probability of 60%" it might be clearer to say, "the probability of cost-effectiveness was 60%."

Response: Thanks for your comment. It was revised based on your comment.

Page 10 – Introduction:

I think in the introduction, there is too much focus on the clinical aspects and too little on the cost-effectiveness. The authors need to explain what we already know about the cost-effectiveness of CBA and RFA then tell us about the limitations of existing economic models and why a new model was needed.

Response: Thanks for your comment. It was revised based on your comment. We explain results of some similar studies in “Introduction” section. In Iran, as mentioned in the text, no full economic evaluation study has been conducted to compare these two technologies, and present study is the first study in Iran in this regard. Due to the fact that the results of similar economic evaluation studies are different in different contexts, the present study was conducted for the first time in Iran. 

Line 73. It is hard to follow this sentence so, I suggest dividing into two or three sentences.

Response: It was revised based on your comment.

Page 11

Line 115. Please, here and everywhere in the text, add "paroxysmal" to be clear

Response: Thanks for your comment. It was revised based on your comment.

Page 12

Line 121. This guideline is not for health projects but for health technologies.

Response: Thanks for your comment. It was revised.

Line 124. I think the authors refer to the design of the economic model - I'd update this sentence as “The economic model was designed based on a literature review ...”

Response: Thanks. It was revised based on your comment.

Line 130. The word “observed” is not meaningful here it should be replaced with a word like “demonstrated”.

Response: Thanks for your comment. It was revised based on your comment.

Line 132. For consistency, please consider updating the first state's name to "before intervention" or the word "intervention" in the other states to "ablation".

Response: Thanks for your comment. It was revised based on your comment.

Lines 136 and 138. Please re-phrase this sentence to be clear and I'd suggest starting it with "It was assumed that..."

Response: Thanks for your comment. Sentences revised based on your comment.

Page 13

Figure 1. I think the model demonstration in Figure 1 doesn't match the description in the text.

Response: Description Sentences was revised. Figure 1 and related description was matched. 

I think the authors should draw another tree. For example, one of the health states is Atrial Fibrillation I think this should be pre-intervention. Similarly, it is confusing to see CBA just after Atrial Fibrillation but not RFA.

Response: Thanks for your comment. Given that the structure of the model is similar in the two comparison arms, in Figure 1 we only included the model structure of one arm. Also, we revised the health state name from “Atrial Fibrillation” to Pre-intervention Atrial fibrillation. Please See revised Fig 1.

In the methods section, the authors need to explain and justify why specific complications were chosen and not the others.

Additionally, the price year needs to be stated in the methods section.

Response: It was revised based on your comment. As mentioned in the text (method section) we included “pericardial effusion” , “cardiac tamponade”, “permanent phrenic nerve palsy”, “vascular complications” and “Stroke” as side effects of the technologies in the model, which we considered both their related cost and the disutility values. The choice of side effect in present study was based on the evidence from previous studies and also in consultation with the clinical specialists.

Line 139. Evidence is a singular word: we cannot write evidence. Please correct this everywhere

Response: It was revised based on your comment.

Line 140. This needs to be explained more which hospital? The hospital that you took the cost data. You may also cite the existing studies here. Additionally, please rephrase this sentence.

Response: Thanks for your comment. It was revised based on your comment and highlighted.

Lines 152 and 153. I don't understand this sentence: could you please rephrase?

Response: It was revised.

Line 163. This needs to be supported with published evidence.

Response: Thanks for your comment. It was revised based on your comment. 

Line 168. Is it publicly available? If so, please provide a reference or provide a summary table as an Appendix.

Response: Thanks for your comment. All the parameters, values and evidence used are given in Table 1. 

Table 1. The authors state following the NICE recommendations; however, they used a discount rate of 0.05. This should be explained and justified. 

Response: We used the NICE reference case to conduct economic evaluations. Only regarding the discount rate, considering the context of Iran, we decided to use a higher discount rate. However, we considered lower and higher values in the sensitivity analysis.

Table 1 includes mortality during first year, but it is not clear what probability was used after the first year. The authors need to explain this in the text.

Response: Thanks for your comment. As you know, the probability of mortality increases with age and this issue is considered in Markov model using table. In Table 1, we only mentioned the probability of death in the first year. We can include the entire probability of death table (Iran Life Table) as S1 in Supporting Information. Please see S1 in Supporting Information.

Table 1 reports the probabilities for complications under the title of “transition probabilities” and this is really confusing. I think Table 1 needs to be re-organised such that the transition probabilities and morbidity risks are clear.

Response: Table 1 revised based on your comment.

The SD or CI should be provided for all the probabilities

Response: Thanks for your comment. We provide SD or CI for variables which were used in sensitivity analysis only. SD or CI for other variables were not used for other variables in the model. 

What is a life table? Please cite a reference or provide this table as an Appendix.

Response: Thanks for your comment. As mentioned above, we include the entire probability of death of normal population (Iran Life Table) as S1 in Supporting Information. Please see S1 in Supporting Information. Also, reference was added in table 1.

In Table 1 some parameters were identified through “survey and calibration” this needs to be explained in the text. What do you mean when you say survey and calibration? We need to see the original figures/values and those used in the model after adjustments.

Response: Thanks for your comment. By “survey and calibration” we mean that the values are not used from another reference and are obtained in the present study. But to avoid misunderstandings we replaced it with “our study” in table 1.

Page 16

Line 189. Citation needed here. I think WHO recommends using three times of GDP as the threshold. Please explain why you have chosen to use the GDP. Please see this paper on the thresholds recommended by WHO https://www.who.int/bulletin/volumes/93/2/14-138206.pdf

Response: Thanks. We added reference. WHO recommended 1 to 3 times the per capita GDP for cost effectiveness threshold in developing countries. 

Also, the following articles have reviewed the threshold recommended by who, saying that three times the per capita GDP as a cost-effectiveness threshold is high. Please see:

1. Bertram MY, Lauer JA, De Joncheere K, et al. Cost-effectiveness thresholds: pros and cons. Bull World Health Organ. 2016;94(12):925–930.

2. Marseille E, Larson B, Kazi DS, Kahn JG, Rosen S. Thresholds for the cost–effectiveness of interventions: alternative approaches. Bull World Health Organ. 2015;93(2):118–124.

Line 200. This needs to be more specific. 10%, 20% or a range of values between 10% and 20%?

Response: As mentioned in manuscript about PSA, in cases which no evidences regarding variance of the variable were found, 10-20% of mean values of variable was considered as standard deviation. In some cases 10% and in some cases 20%.

Line 201. Please provide a reference here to show how you have chosen the appropriate distributions.

Response: Thanks for your comment. As mentioned in manuscript, the range used for uncertainty, statistical distributions and related reference of each variable used in PSA are presented in table 1. In cases which no evidences regarding variance of the variable were found, 10-20% of mean values of variable was considered as standard deviation. In some cases 10% and in some cases 20%.

Page 17

Line 204. Please use commas for costs to make sure these are clear. For example, is it $14,198.3?

Response: Thanks for your comment. It was revised based on your comment.

Table 2. Please re-organise this table and report the incremental figures in a separate raw. Also, please change the column name from Eff** to QALYs to be more specific.

Response: It was revised based on your comment. Please see revised Table 2.

Page 18

Figure 3. This figure is confusing because the impact on the net benefit was demonstrated which was not mentioned anywhere in the text. Please repeat this analysis to show the impact of changing inputs on ICER calculations. Additionally, we cannot tell which lower and upper values were used. So, please provide these on the graph. Please re-label the parameters and try not to use abbreviations. Please

also explain what EV represents. Again, use commas for costs.

Response: Thanks for your comment. It was revised based on your comment. Also, we mentioned EV (Expected Value) represents ICER. Please see revised figure 3. Also, commas were used in whole text for costs. 

Figures 4 and 5. Please repeat these analyses using ICERs not net benefits.

Response: Thanks for your comment. We repeat the sensitivity analysis again based on your comment. Please see revised figure 3.

Line 242. Why 1000 times? why not 5000 or 10000? This needs to be justified. 

Response: Thanks for your comment. As you know, there is no specific reference to how many times we do the sampling in monte-carlo simulation, and researchers usually determine this amount based on the study, which is usually considered 100 or 1000 times in most studies. But in order to be more precise in this regard, we repeat the monte-carlo simulation by 10,000 times sampling. Please see.

Line 243. This sentence raises concern. It was said that the cost-effectiveness of CBA would be estimated compared to RFA. However, the PSA was conducted to estimate the cost-effectiveness of CBA and RFA compared to something else which was not explained to the reader. Is the probability of CBA being cost-effective compared to RFA 59% as stated in the abstract or is the probability of CBA being cost-effective compared to something else is 59%? The authors need to clarify this and repeat the analysis to compare CBA to RFA.

Response: We definitely meant that the probability of CBA being cost-effective compared to RFA is 59% based on strategy selection diagram. Of course, we deleted this chart based on reviewers’ comment at this stage. We reported the PSA findings solely on the basis of the Scatter plot and 10,000 iterations.

Page 19

Figure 7. This figure is in line with the text, so maybe use this on and remove Figure 6 and related text. Here, please change effectiveness to QALYs.

Response: It was revised based on your comment.

Table 3. I don't think this makes sense. We don't need this table, just remove it please.

Response: Thanks for your comment. We reported this table to determine the exact percentage placement of interventions in each quadrant of the cost-effectiveness plane. However, if you insist on this comment, we can delete this section in the next step.

Page 20

Lines 262 and 265. This part should be moved to introduction.

Response: Thanks. We would like this section not to be removed from the discussion. In fact, we used this as an introduction to this section to compare the effectiveness and cost-effectiveness of technologies. However, if you insist on this comment, we can delete this section in the next step.

Line 269. Please don't repeat the results here - it would be enough just to say that it was not cost-effective at a WTP of $7,142 but cost-effective at a WTP of $14,285.

Response: Sentences was revised based on your comment.

Line 279. Why would the cost of RFA increase? Please rephrase this as "the cost of RFA is $5,738 not $5,127 ..."

Response: We rephrased it. Please see.

Line 282. I don't understand this phrase (re-use of necessities). Please explain what this means.

Response: We revised this phrase. Please see discussion.

Page 21

Line 298. We wouldn't say viewpoint to describe the perspective in economic evaluations.

Response: Thanks for your comment. It was revised based on your comment.

Reviewer #2:

Thank you for your invaluable comments and guidance to improve our manuscript. 

We presented the responses one by one on your comments as follows: 

Please remove "Case Study" from the title.

Response: Thanks for your comment. It was revised based on your comment.

 English language of the manuscript needs to be improved. 

Response: Thanks for your comment. Manuscript was revised by native English editor.

Proof-editing the manuscript because grammar, punctuation and use of acronyms is rather poor. 

Response: Thanks for your comment. Manuscript was revised by native English editor.

Please check keywords with Mesh terms. 

Response: It was revised based on your comment.

The authors should harmonize the cost-utility/ cost utility in the manuscript. 

Response: Thanks for your comment. Manuscript was revised based on your comment.

Lines 188-189: please add some references. 

Response: Thanks for your comment. Reference was added.

Line 190: TreeAge 2011 software/ line 59: TreeAge pro 2020 software. Please correct.

Response: Thanks for your comment. It was revised. We used TreeAge pro 2020 software.

---

## [Decision Letter · Decision Letter 1]

16 May 2022

PONE-D-21-27154R1Cost-Utility Analysis of Cryoballoon Ablation versus Radiofrequency Ablation in the Treatment of Paroxysmal Atrial Fibrillation in IranPLOS ONE

Dear Dr. Daroudi,

Thank you for submitting your manuscript to PLOS ONE. After careful consideration, we feel that it has merit but does not fully meet PLOS ONE’s publication criteria as it currently stands. Therefore, we invite you to submit a revised version of the manuscript that addresses the points raised during the review process.

We look forward to receiving your revised manuscript.

Kind regards,

Elena Pizzo, PhD

Academic Editor

PLOS ONE

Journal Requirements:

Additional Editor Comments:

Dear authors,

Thank you for submitting your revised version.

The paper has been improved substantially but some additional minor revisions are required.

Please address the reviewers' comments below.

Reviewers' comments:

Reviewer's Responses to Questions

**Comments to the Author**

1. If the authors have adequately addressed your comments raised in a previous round of review and you feel that this manuscript is now acceptable for publication, you may indicate that here to bypass the “Comments to the Author” section, enter your conflict of interest statement in the “Confidential to Editor” section, and submit your "Accept" recommendation.

Reviewer #1: (No Response)

Reviewer #2: All comments have been addressed

2. Is the manuscript technically sound, and do the data support the conclusions?

Reviewer #1: Yes

Reviewer #2: Yes

3. Has the statistical analysis been performed appropriately and rigorously? 

Reviewer #1: Yes

Reviewer #2: Yes

4. Have the authors made all data underlying the findings in their manuscript fully available?

Reviewer #1: Yes

Reviewer #2: Yes

5. Is the manuscript presented in an intelligible fashion and written in standard English?

Reviewer #1: Yes

Reviewer #2: Yes

6. Review Comments to the Author

Reviewer #1: General comments

I thank the authors for their work, and I think the manuscript has been improved significantly. I have some additional comments.

A final, through proof-reading is needed since there are some grammatical errors (for example tense inconsistencies) and several sentences that could be improved. Add page numbers, please.

Introduction

Line 74 – Please replace the word “pass” with “follow”.

Line 106 – Please replace the word “various” with “some”.

Line 116 – Please rephrase this sentence as follows: “Because CBA is a relatively new technology in Iran, …”

Figure 1 – Please make a note at the bottom of the Figure to state that the structure is the same for the control arm.

Methods

Line 152 – Please write QALY in full here since it is the first time that it is been used. So, it should be: “quality adjusted life years (QALYs)”

Please write the discount rate here and explain why you used 5% instead of 3% as suggested by NICE. Is there a guideline in Iran which suggests that 5% should be used?

The price year must be provided in the methods section. Are these 2021 $?

Please provide the “uncalibrated” figures and the “calibrated” figures. This way the reader can tell if the authors’ calibrations were correct. I’d call them adjustments rather than calibrations.

Line 197 – I think this sentence is abundant given that the next sentence clearly explains which WTP was used.

Line 209 – Please provide a reference for the choice of distributions (i.e. beta and gamma).

Line 211 – Please replace “10-20%” with “10% or 20%” for clarity.

Figure 3 – Please refrain from using EV instead of ICER because it might be confused with another type of analysis (expected value of information). Please only use ICER.

Discussion

Most of the clinical and the cost data came from a study conducted in China. I think this is an important limitation and should be acknowledged. Please discuss the potential impacts of this. How can decision-makers be sure that these results are applicable to Iran? Please also discuss the potential impact on health inequalities.

Reviewer #2: (No Response)

7. PLOS authors have the option to publish the peer review history of their article (what does this mean?). If published, this will include your full peer review and any attached files.

Reviewer #1: No

Reviewer #2: No

---

## [Author Response · Author response to Decision Letter 1]

23 May 2022

Response to reviewers: Round 2

Reviewer 1 Comments:

Dear reviewer

Thank you for your invaluable comments and guidance to improve our manuscript. 

We presented the responses one by one on your comments as follows: 

Reviewer #1: General comments:

I thank the authors for their work, and I think the manuscript has been improved significantly. I have some additional comments.

A final, through proof-reading is needed since there are some grammatical errors (for example tense inconsistencies) and several sentences that could be improved. Add page numbers, please.

Response: Thanks for your comment. Another proof-reading was conducted.

Introduction

Line 74 – Please replace the word “pass” with “follow”.

Response: It was revised based on your comment.

Line 106 – Please replace the word “various” with “some”.

Response: It was revised based on your comment.

Line 116 – Please rephrase this sentence as follows: “Because CBA is a relatively new technology in Iran, …” 

Response: Thanks for your comment. It was revised based on your comment.

Figure 1 – Please make a note at the bottom of the Figure to state that the structure is the same for the control arm. 

Response: Thanks. It was revised based on your comment. 

Line 152 – Please write QALY in full here since it is the first time that it is been used. So, it should be: “quality adjusted life years (QALYs)”

Response: Thanks. It was revised based on your comment.

Please write the discount rate here and explain why you used 5% instead of 3% as suggested by NICE. Is there a guideline in Iran which suggests that 5% should be used?

Response: We did not have a guideline for this purpose in Iran until the study. We used 5% as discount rate for both costs and QALYs in the model based on the recommendation of the Health Technology Assessment Office of the Iran’s Ministry of Health for this study. This descriptions was also added in the Methods section of the manuscript.

The price year must be provided in the methods section. Are these 2021 $?

Response: Thanks for your comment. We mentioned in line 175 of the manuscript that “All costs were calculated based on 2019-20 prices”. Please see.

Please provide the “uncalibrated” figures and the “calibrated” figures. This way the reader can tell if the authors’ calibrations were correct. I’d call them adjustments rather than calibrations.

Response: Thanks for your comment. As mentioned in the previous correspondence, by “survey and calibration” we mean that the values are not used from another reference and are obtained in the present study. But to avoid misunderstandings we replaced it with “our study” in table 1. 

Also, in cases where calibration is included with a reference as a variable source, the reader can see the original value of the variable by referring to the reference. Also, we replaced calibration with adjustment.

Line 197 – I think this sentence is abundant given that the next sentence clearly explains which WTP was used.

Response: It was revised based on your comment.

Line 211 – Please replace “10-20%” with “10% or 20%” for clarity.

Response: Thanks. It was revised based on your comment.

Figure 3 – Please refrain from using EV instead of ICER because it might be confused with another type of analysis (expected value of information). Please only use ICER.

Response: It was revised based on your comment.

Discussion

Most of the clinical and the cost data came from a study conducted in China. I think this is an important limitation and should be acknowledged. Please discuss the potential impacts of this. How can decision-makers be sure that these results are applicable to Iran? Please also discuss the potential impact on health inequalities.

Response: Thanks for your comment. As you know, the use of clinical evidence from past studies is very common in economic evaluation studies and is not usually considered as a limitation. In present study we used the best present evidence of international studies for clinical parameters, and for cost items we used internal evidence. As mentioned in the text, although in many cases there is no significant difference between different contexts regarding clinical evidence, but this is one of the possible limitations of this study that can be effective in final results. However, to eliminate this possible limitation as much as possible, in present study we performed sensitivity analysis to a wide range for uncertain parameters.

---

## [Decision Letter · Decision Letter 2]

15 Jun 2022

Cost-Utility Analysis of Cryoballoon Ablation versus Radiofrequency Ablation in the Treatment of Paroxysmal Atrial Fibrillation in Iran

PONE-D-21-27154R2

Dear Dr. Daroudi,

We’re pleased to inform you that your manuscript has been judged scientifically suitable for publication and will be formally accepted for publication once it meets all outstanding technical requirements.

Kind regards,

Elena Pizzo, PhD

Academic Editor

PLOS ONE

Reviewers' comments:

Reviewer's Responses to Questions

**Comments to the Author**

1. If the authors have adequately addressed your comments raised in a previous round of review and you feel that this manuscript is now acceptable for publication, you may indicate that here to bypass the “Comments to the Author” section, enter your conflict of interest statement in the “Confidential to Editor” section, and submit your "Accept" recommendation.

Reviewer #1: All comments have been addressed

2. Is the manuscript technically sound, and do the data support the conclusions?

Reviewer #1: Yes

3. Has the statistical analysis been performed appropriately and rigorously? 

Reviewer #1: Yes

4. Have the authors made all data underlying the findings in their manuscript fully available?

Reviewer #1: Yes

5. Is the manuscript presented in an intelligible fashion and written in standard English?

Reviewer #1: Yes

6. Review Comments to the Author

Reviewer #1: I would like to thank the authors for their valuable work.

I do not have any further comments.

Best wishes

7. PLOS authors have the option to publish the peer review history of their article (what does this mean?). If published, this will include your full peer review and any attached files.

Reviewer #1: No

---

## [Editor Report · Acceptance letter]

21 Jun 2022

PONE-D-21-27154R2 

Cost-Utility Analysis of Cryoballoon Ablation versus Radiofrequency Ablation in the Treatment of Paroxysmal Atrial Fibrillation in Iran 

Dear Dr. Daroudi:

I'm pleased to inform you that your manuscript has been deemed suitable for publication in PLOS ONE. Congratulations! Your manuscript is now with our production department. 

Kind regards, 

on behalf of

Dr. Elena Pizzo 

Academic Editor

PLOS ONE